# In Vitro Evaluation of the Inhibitory Activity of Different Selenium Chemical Forms on the Growth of a *Fusarium proliferatum* Strain Isolated from Rice Seedlings

**DOI:** 10.3390/plants10081725

**Published:** 2021-08-20

**Authors:** Elisabetta Troni, Giovanni Beccari, Roberto D’Amato, Francesco Tini, David Baldo, Maria Teresa Senatore, Gian Maria Beone, Maria Chiara Fontanella, Antonio Prodi, Daniela Businelli, Lorenzo Covarelli

**Affiliations:** 1Department of Agricultural, Food and Environmental Sciences, University of Perugia, 06121 Perugia, Italy; elisabetta8891@gmail.com (E.T.); francesco.tini@collaboratori.unipg.it (F.T.); daniela.businelli@unipg.it (D.B.); lorenzo.covarelli@unipg.it (L.C.); 2Department of Agricultural and Food Sciences, Alma Mater Studiorum University of Bologna, 40127 Bologna, Italy; david.baldo@unibo.it (D.B.); mariateresa.senatore@unibo.it (M.T.S.); antonio.prodi@unibo.it (A.P.); 3Department for Sustainable Food Process, Catholic University of the Sacred Heart of Piacenza, 29122 Piacenza, Italy; gian.beone@unicatt.it (G.M.B.); mariachiara.fontanella@unicatt.it (M.C.F.)

**Keywords:** fungi, *Fusarium*, selenium, micronutrient, inhibition, bioactivity

## Abstract

In this study, the in vitro effects of different Se concentrations (5, 10, 15, 20, and 100 mg kg^−1^) from different Se forms (sodium selenite, sodium selenate, selenomethionine, and selenocystine) on the development of a *Fusarium proliferatum* strain isolated from rice were investigated. A concentration-dependent effect was detected. Se reduced fungal growth starting from 10 mg kg^−1^ and increasing the concentration (15, 20, and 100 mg kg^−1^) enhanced the inhibitory effect. Se bioactivity was also chemical form dependent. Selenocystine was found to be the most effective at the lowest concentration (5 mg kg^−1^). Complete growth inhibition was observed at 20 mg kg^−1^ of Se from selenite, selenomethionine, and selenocystine. Se speciation analysis revealed that fungus was able to change the Se speciation when the lowest Se concentration was applied. Scanning Electron Microscopy showed an alteration of the fungal morphology induced by Se. Considering that the inorganic forms have a higher solubility in water and are cheaper than organic forms, 20 mg kg^−1^ of Se from selenite can be suggested as the best combination suitable to inhibit *F. proliferatum* strain. The addition of low concentrations of Se from selenite to conventional fungicides may be a promising alternative approach for the control of *Fusarium* species.

## 1. Introduction

Selenium (Se) is an essential micronutrient for humans and animals, and is involved in numerous biological processes, such as cellular response to oxidative stress, cellular differentiation, redox signaling, and protein folding [1,2,3]. More than 25 Se-containing proteins have been identified in mammals, having a role in the regulation of redox processes. Among Se proteins, Se is a crucial component of glutathione peroxidase, whose main biological role is to protect against oxidative damage by reducing free hydrogen peroxide to water and lipid hydroperoxides to their corresponding alcohols [4].

In addition to humans and animals, Se is also beneficial to plants when applied at low concentrations [5,6,7,8]. For example, Se contributes to the control of water status [9], prevents oxidative stress, delays senescence, and promotes growth [10]. Due to this experimental evidence, numerous studies have investigated Se-biofortification strategies for providing plant protection against abiotic stresses and, at the same time, when possible, beneficial food for human health [11,12,13]. As demonstrated for many other nutrients, Se also shows a U-shaped relationship between concentration in living organisms and the risk of deficiency toxicity that occurs both below and above the physiological range, which is very narrow [3,14]. Furthermore, Se effects on living organisms are not only concentration dependent but are also related to its chemical form and its bioavailability [15].

Depending on certain experimental conditions (i.e., application mode, concentration, form, and application timing), Se is beneficial to plants and, at the same time, detrimental to plant pathogens [16,17,18]. For this reason, the activity and role of Se within a plant–pathogen interaction are worthy of more in-depth elucidation. To date, several studies concerning the use of Se salt treatment for the control of a range of plant pathogens have been undertaken, such as, *Aspergillus funiculosus*, *Alternaria tenuis*, *Fusarium* spp. and *Fusarium graminearum* in artificial media [19,20,21]; *Fusarium* spp. and *Alternaria brassicicola* in Indian mustard [22]; *Fusarium oxysporum* f. sp. *lycopersici* in tomato [23]; *Penicillium expansum* in artificial media [18]; *Botrytis cinerea* in tomato [24]; and *F. graminearum* in wheat [21]. Other studies showed the Se protective effect in plants against the activity of mycotoxins, such as zearalenone [25,26] and aflatoxin B_1_ [27]. The in vitro and in vivo ability of Se to reduce the deoxynivalenol (DON) production by *F. graminearum* was also explored [21]. Additionally, in soil, Se was found to enhance the microbiome diversities and the relative abundance of Plant Growth Promoting Bacteria (PGPB), while reducing the number of pathogenic fungi [28].

These results indicate that Se might serve as a potential alternative to synthetic fungicides for the control of certain plant diseases caused by several fungal pathogens [18].

In a previous study [29] on rice (*Oryza sativa* L.) seedlings (variety Selenium) cultivated in a hydroponic system in the presence of half-strength Hoagland solution [30], browning of stem bases was detected in more than 50% of total plants. Interestingly, the rice seedlings grown with the same method but in the presence of a Se salt (sodium selenite at a concentration of 20 mg L^−1^ of Se) showed a noticeably lower presence of these symptoms (observed only in approximately 5% of total plants). This observation, as successively described in the present paper, led us to identify the fungal microorganism associated with these symptoms as belonging to the species *Fusarium proliferatum*, a member of the *Fusarium fujikuroi* species complex (FFSC), a group of 40 closely related *Fusarium* species defined by morphological traits, sexual compatibility, and DNA-based phylogenetic analysis [31,32,33]. *F. proliferatum* is a globally widespread causal agent of diseases of various economically important plants including staple crops such as cereals. *F. proliferatum* is mostly found to colonize maize [34], but has also been isolated from rice [35,36], wheat [33], sorghum, millet [37,38], asparagus [39], garlic [40], and date palm [41]. In rice, *F. proliferatum* is a well-known pathogen associated with Bakanea disease. This seed-borne disease is also caused, in addition to *F. proliferatum*, by other species belonging to FFSC, such as *Fusarium fujikuroi* and *Fusarium verticillioides*. The typical symptoms of Bakanae disease are seedling blight, root and crown rot, pale green to yellowing of foliage, chlorotic leaves, and abnormal elongation [36,42,43].

The presence of *F. proliferatum* in plants is also a potential risk to animal and human health because of its ability to biosynthesize several mycotoxins, such as fumonisins [44]. In particular, maize and, to a lesser extent, rice, are the matrices in which natural contaminations of this mycotoxin are more common [45].

Due to the global importance of both *F. proliferatum* and the host (rice) from which it has been isolated, and because, to the best of our knowledge, no studies have been conducted on the effect of Se against *F. proliferatum*, in the present study we investigated the in vitro effect of various Se concentrations (5, 10, 15, 20, and 100 mg kg^−1^ of Se) from four different Se chemical forms (sodium selenite, sodium selenate, selenomethionine (Se-Met), and selenocystine (Se-Cys)) on the development of a *F. proliferatum* strain isolated from rice seedlings. Additionally, a Se speciation analysis was performed to obtain information on the occurrence of Se metabolites and their distribution, as a consequence of Se bioconversion operated by the fungus. Finally, Scanning Electron Microscopy (SEM) analysis was also carried out to investigate the possible modifications induced by Se on the hyphal morphology of *F. proliferatum*.

As a result, the best combination of Se form and Se concentration suitable for the in vitro inhibition of the development of *F. proliferatum* is described. In addition, the obtained results were found to be useful for the hypothesis of new and alternative approaches to manage *F. proliferatum* infections of rice and other cereal crops, perhaps coupled with a Se-biofortification strategy [46] of grains destined for human and animal food products.

## 2. Results

### 2.1. Identification of the F. proliferatum Strain PG-CH1 Isolated from Rice Seedlings

The strain PG-CH1, obtained from rice seedlings, was identified as *F. proliferatum* showing a similarity score of >99% with the reference sequences of the same species deposited on NCBI and *Fusarium MLST* databases. The identification was also confirmed by phylogenetic analysis (Figure 1) based on a single locus dataset of *translation elongation factor 1α* (*tef1α*) partial sequences (631 bp). The strain PG-CH1 clustered together with the reference strain *F. proliferatum* G18SXS9-2 (Accession Number MK952837) and *Gibberella intermedia* S1S (sexual stage of *F. proliferatum*; Accession Number JN092349) showed high bootstrap support (95%).

### 2.2. The In Vitro Inhibitory Activity of Different Se Forms on F. proliferatum Strain PG-CH1 Growth

One representative image for each treatment (untreated or treated with Se from different forms and concentrations) is shown in Figure 2, in which the in vitro inhibitory activity of Se on *F. proliferatum* strain PG-CH1 colony growth on Potato Dextrose Agar (PDA) is reported. In combination with the growth reduction, as can be inferred from Figure 2, an alteration induced by Se on the morphology of *F. proliferatum* strain PG-CH1 colonies was also observed. In particular, a lower cottony texture and a lower mycelium density, in addition to a more intense red pigmentation in the central area of the colony, were visible in the presence of Se in comparison to the untreated control. These morphological modifications were macroscopically more appreciable in the presence of lower Se concentrations (5–10 mg kg^−1^) that allowed a certain colony development. Interestingly, the presence of Se from selenate induced a considerable mycelium ramification (Figure 2b) that was not detected in the presence of Se from other chemical forms. To improve clarity, please note that the ramification was considered for radial growth measurement. Data relative to radial growth reduction in *F. proliferatum* strain PG-CH1 colonies concerning the untreated control (0 mg kg^−1^ of Se), measured after 10 days of incubation at 22 ± 2 °C in the dark in the presence of various Se concentrations (5, 10, 15, 20, and 100 mg kg^−1^) from different Se forms (selenite, selenate, Se-Met, and Se-Cys), are summarized in Figure 3.

Significant differences in radial growth reduction in *F. proliferatum* strain PG-CH1 colonies relative to the untreated control (0 mg kg^−1^ of Se) were detected both within a Se form for different concentrations and within a Se concentration for different Se forms. Focusing the attention on Se from selenite (Figure 3), the inhibitory activity of 5 mg kg^−1^ concentration was significantly lower (*p* < 0.05) than those caused by all other concentrations, whereas Se concentrations of 15, 20, and 100 mg kg^−1^ showed a significantly higher inhibitory activity (*p* < 0.05) than the concentration of 10 mg kg^−1^. The same significant (*p* < 0.05) gradient described for selenite was observed for Se-Met (Figure 3). In contrast, regarding Se from selenate (Figure 3), the inhibitory activity caused by Se concentrations of 5, 10, 15, and 20 mg kg^−1^ was not significantly different (*p* > 0.05). Only the highest Se concentration (100 mg kg^−1^) showed a significant effect (*p* < 0.05) on the reduction in *F. proliferatum* strain PG-CH1 growth in comparison to the inhibition caused by all other concentrations.

The growth reduction performed by Se from Se-Cys (Figure 3) was not significantly different (*p* < 0.05) across the five Se concentrations tested; that is, the inhibitory activity detected at the lowest concentration of 5 mg kg^−1^ of Se was not different from that observed following the treatment with higher Se concentrations, including that of 100 mg kg^−1^. The concentration of 5 mg kg^−1^ Se from Se-Cys showed a significantly higher (*p* < 0.05) inhibitory activity with respect to Se from all other chemical forms. Increasing the concentration to 10 mg kg^−1^ of Se from Se-Cys and selenite showed a similar (*p* > 0.05) effect on the growth reduction in *F. proliferatum* strain PG-CH1, followed by Se from Se-Met. Se inhibition activity from this last compound was, in turn, significantly lower (*p* < 0.05) than Se from Se-Cys but not significantly different (*p* > 0.05) from that observed for Se from selenite. Conversely, at this concentration, Se from selenate showed the significantly lowest (*p* < 0.05) inhibitory activity compared with that of Se from the other three forms. At the concentrations of 15 and 20 mg kg^−1^, Se from selenite, Se-Cys, and Se-Met showed a similar (*p* > 0.05) effect on fungal growth reduction, whereas Se from selenite showed, in both cases, a significantly lower (*p* < 0.05) inhibitory activity than those detected for Se from the other three chemical forms. Finally, at the highest tested concentration (100 mg kg^−1^), the inhibitory activity of Se from Se-Cys was significantly lower (*p* < 0.05) than those detected for Se from selenite, selenate, and Se-Met.

Summarizing, all Se treatments from different Se forms showed a certain effect on *F. proliferatum* strain PG-CH1. In particular, the highest inhibitory activity at the lowest concentration of Se (5 mg kg^−1^) tested in this experiment was caused by Se from Se-Cys, whereas increasing the concentrations (10, 15, and 20 mg kg^−1^) of Se from selenite, Se-Cys, and Se-Met showed a higher reduction in fungal colony growth than Se from selenate. Se from selenite, selenate, and Se-Met inhibited *F. proliferatum* strain PG-CH1 growth more than Se-Cys at 100 mg kg^−1^. The control treatment (PDA amended with Na^+^ 100 mg kg^−1^ from Sodium chloride (NaCl)) showed a negligible effect on the growth of *F. proliferatum* strain PG-CH1 colonies, because only a radial growth reduction up to 1.8% was observed (Appendix A).

### 2.3. Se Speciation

Se speciation analysis was performed on Se-amended PDA in the presence of *F. proliferatum* strain PG-CH1, and on Se-amended PDA in the absence of *F. proliferatum* strain PG-CH1. In the presence of fungus, Se-speciation data showed a transformation of the applied Se form (selenite, selenate, Se-Cys, and Se-Met) in other Se chemical forms (Figure 4). Conversely, no transformation of the applied Se form was observed in the absence of *F. proliferatum* strain PG-CH1 (data not reported).

In general, in the presence of fungus, the observed Se transformation in other Se chemical forms was reduced by increasing Se concentration to such an extent that no conversion was detected at the highest Se concentration (100 mg kg^−1^). This observation suggests the capacity of the fungus to metabolize Se at low concentrations, whereas accumulation of the applied Se form occurred at the highest Se concentration. For that reason, in the following, we discuss the Se speciation results in the presence of *F. proliferatum* strain PG-CH1, focusing on the lowest Se concentrations applied of 5 and 10 mg kg^−1^. When inorganic Se from selenite was applied, the conversions in Se-Cys (up to 36%) and Se-Met (up to 16%) were mainly observed (Figure 4a); when inorganic Se from selenate was applied, the conversions in Se-Met (up to 58%), Se-Cys (up to 16%), and selenite (up to 8%) occurred (Figure 4b). When organic Se from Se-Met was applied, Se speciation analysis revealed mainly the conversion in Se-Cys (up to 72%) and a trace of selenite (up to 4%) (Figure 4c); when Se-Cys was applied, it was converted to selenite (up to 24%), selenate (up to 11%), and Se-Met (up to 8%) (Figure 4d).

To ascribe with certainty the Se transformation in other Se forms to *F. proliferatum* strain PG-CH1, Se speciation was also performed on PDA amended with Se in the absence of the fungus. Based on the above results (Figure 4), the lowest Se concentration (as the concentration at which the greater Se conversion into other Se forms occurred) was selected and Se speciation analysis was performed on PDA amended with 5 mg kg^−1^ of Se from selenite, selenate, Se-Cys, and Se-Met in the absence of *F. proliferatum* strain PG-CH1. The obtained data did not show the transformation of the applied Se form in other Se chemical forms, thus confirming that the observed Se transformations in other chemical forms were undertaken by *F. proliferatum* strain PG-CH1.

### 2.4. Selenium (20 mg kg^−1^) from Sodium Selenite Effect Observed by SEM

Based on the results obtained for the in vitro inhibitory activity of Se from different Se forms on *F. proliferatum* strain PG-CH1 growth, for SEM analysis we focused our attention on the effect of 20 mg kg^−1^ of Se from selenite (Figure 5).

In general, a lower mycelium density with shorten hyphae was observed in the treated samples (Figure 5d) in comparison to the untreated ones (Figure 5a). In detail, a strong hyphal collapse was also noticed in the treated samples (Figure 5e,f) compared to the untreated samples (Figure 5b,c). This observation showed additional evidence that Se from selenite at the concentration of 20 mg kg^−1^ was able to determine the inhibition of *F. proliferatum* strain PG-CH1 growth, which was also manifested by the alteration of fungal hyphae morphology and density.

## 3. Discussion

This study aimed to assess the inhibitory activity of Se toward *F. proliferatum* strain PG-CH1 isolated from rice seedlings. Several studies [3,14,15] showed that Se effects on living organisms depend on Se concentration, in addition to the Se chemical form and bioavailability. Se is one of the few non-metals exhibiting variable oxidation states—selenate (+6), selenite (+4), elemental Se (0), and selenide (−2). Moreover, the distribution of the valence states in a given environment strongly depends on several parameters, such as biological interactions, pH, redox conditions, the solubility of its salts, the chemical complex of soluble and solid ligands, reaction kinetics, and temperature [47,48]. Consequently, Se naturally occurs in different chemical forms. Among these, in living organisms, Se is found both in inorganic forms (mainly as selenite and selenate) and organic forms (mainly as Se-Cys and Se-Met) [49]. In general, it has been reported that organic Se forms have less toxicity compared to inorganic forms in living organisms [3].

Based on the above, in this study, four different Se chemical forms (inorganic Se as selenite and selenate; organic Se as Se-Met and Se-Cys) in the concentration range 5–100 mg kg^−1^ of Se were tested. This concentration range was selected after a preliminary test in which *F. proliferatum* strain PG-CH1 was grown on PDA amended with Se concentration (from selenite) in the range 1–200 mg kg^−1^. A negligible effect on the reduction in the fungal growth was observed up to 5 mg kg^−1^ of Se, whereas the fungal growth was almost completely inhibited at 100 mg kg^−1^ of Se.

In general, in our experimental conditions, a significant in vitro concentration-dependent Se bioactivity from different Se forms towards *F. proliferatum* strain PG-CH1 was observed. For example, with the exception of Se from selenate, Se reduced (>60%) the *F. proliferatum* strain PG-CH1 growth relative to the untreated control starting from the concentration of 10 mg kg^−1^. Additionally, by increasing Se concentration to 15 and 20 mg kg^−1^, the growth reduction caused by Se from selenite, Se-Met, and Se-Cys exceeded 80%. Finally, at the concentration of 100 mg kg^−1^ of Se, *F. proliferatum* strain PG-CH1growth was almost completely inhibited (growth reduction of ≥89%), regardless of the applied Se form. An in vitro concentration-dependent activity of Se was previously reported to inhibit *A. funiculosis*, *A. tenuis*, and *Fusarium* spp. (Se from selenite) [19], *P. expansum* (Se from selenite) [18], *B. cinerea* (Se from selenite) [50], *A. brassicicola*, *Fusarium* spp. (Se from selenate) [22], and *F. graminearum* (Se from selenite, selenate, Se-Met, and Se-Cys) [21] growth. In addition to reporting that the effect of Se (from selenite) against a strain of *Fusarium* sp. was concentration dependent, Yin et al. 2017 [51], showed that Se (from selenite) in the range 0.1–1.0 mg kg^−1^ promoted the growth of another strain belonging to the *F. tricinctum* species.

Moreover, the results obtained in this study showed that Se bioactivity was not only concentration dependent but, as mentioned above, also related to the Se chemical form deployed and its bioavailability. For example, Se from selenate exhibited the lowest bioactivity in up to 20 mg kg^−1^ of Se (radial growth reduction in *F. proliferatum* strain PG-CH1 colonies of up to 28%) compared to the other Se forms (percentage of radial growth reduction in *F. proliferatum* colonies ranged from 83 to 97% at 20 mg kg^−1^ of Se from selenite, Se-Cys, and Se-Met). Conversely, Se from Se-Cys was found to be the most effective at the lowest Se concentration (radial growth reduction in *F. proliferatum* colonies at 5 mg Kg^−1^ of 79.3%, whereas Se from other forms reduced the growth by up to 13.1% at 5 mg kg^−1^ of Se). However, it should be recalled that, at the concentrations of 10, 15, and 20 mg kg^−1^, Se from selenate altered the morphology of *F. proliferatum* strain PG-CH1 colonies by inducing hyphae ramification and decreasing mycelium density. In addition, in vitro (20–80 mg L^−1^) and in planta (wheat, 20 mg L^−1^) effects of Se from selenite, selenate, Se-Met, and Se-Cys on fungal growth (in vitro), symptoms (in planta), and DON accumulation (in vitro and in planta) was compound dependent [21].

For further investigation of the process of growth inhibition resulting from Se supplementation to the growth medium, we performed Se speciation analysis to obtain information on the occurrence of the process of Se conversion in other chemical forms operated by *F. proliferatum* strain PG-CH1, or on the process of Se accumulation. No transformation of the applied Se forms occurred in the absence of *F. proliferatum*, whereas transformation of the applied Se forms was observed in the presence of the fungal microorganism, indicating that *F. proliferatum* strain PG-CH1 can metabolize Se.

In particular, the fungal capacity to transform the applied Se form was reduced by increasing Se concentration (Figure 4). Thus, we can hypothesize that the complete growth inhibition in *F. proliferatum* strain PG-CH1 colonies observed at 100 mg kg^−1^ of Se may be related to the accumulation (and consequent toxicity) of the applied Se forms. Concerning the lowest Se concentration applied, the observed inhibition of *F. proliferatum* strain PG-CH1 growth is more difficult to explain. Based on the collected speciation data and considering that the biochemistry of Se resembles that of sulfur (S) [52], we hypothesize several mechanisms as follows:(i)The occurrence of redox processes mediated by Se (e.g., selenate reduction to selenite, see Figure 4b), which may have affected the regular physiological processes of the fungal microorganism.(ii)The formation of organic Se compounds caused by the fungus (e.g., inorganic Se is converted to Se-Met and Se-Cys, see Figure 4a,b; Se-Met is converted to Se-Cys, see Figure 4c; Se-Cys is converted to Se-Met in small amounts, see Figure 4d), which may lead to direct incorporation of Se-Cys and Se-Met into proteins, rather than the analog sulfur-containing amino acids S-Cys and S-Met, a process previously reported in the literature [53,54]. In this case, the growth inhibitory effect may be ascribed to a different conformation of the Se proteins concerning analog S proteins to such an extent that a modification in protein activity should not be excluded.(iii)The direct incorporation of the applied Se-Cys and Se-Met into proteins, a process which may explain the higher bioavailability of organic Se forms compared to inorganic Se forms.(iv)The relationship of Se with oxidative stress [5]. For example, it is known that in certain filamentous fungi, mycelium metamorphosis, in structures that ensure fungal propagation, is induced by increased oxidative stress. Conversely, decreased oxidative stress causes a permanence of undifferentiated mycelia, and inhibition of metamorphosis and fungal propagation [55]. The extension of this theory predicts that any antioxidant (such as Se at certain doses) can stop fungal propagation by inhibiting its metamorphosis, thus acting as a natural fungicide.

To support hypotheses (ii) and (iii) for explaining the Se toxic effect, it has been reported that an excess of Se-containing proteins can have adverse effects on cellular metabolism [56]. Finally, we also noted a strong smell in the culture medium amended with Se, which may be due to the production of the volatile compound dimethyl selenide, (CH_3_)_2_Se, as a consequence of the detoxification mechanism of the fungus. As reported in the literature [57], the methylation of inorganic Se by microorganisms produces volatile inorganic compounds that are less toxic than inorganic forms.

Se speciation analysis is a powerful tool for detecting the final products of the Se bioconversion operated by fungi; nevertheless, Se speciation data alone are not sufficient to provide an in-depth explanation of the metabolic pathway of Se in fungal microorganisms. Several authors attempted to explain the bioavailability and toxicity of the Se forms for fungal microorganisms at the molecular and/or physiological levels. For example, Reference [58] reports that selenite and selenate are assimilated through oxyanion transporters, and, once inside the cells, they are transformed into selenide through a reductive pathway that may involve the enzyme glutathione peroxidase. In the presence of oxygen, selenide can promote the formation of reactive oxygen species (ROS), which may damage DNA, proteins, and other cellular macromolecules. Selenide is also the intermediate for the formation of selenocystine, from which selenomethionine can then be formed in organisms with a functional transsulfuration pathway. Several metabolomic studies demonstrated that selenocysteine can also be formed from selenomethionine [59,60]. However, the reason for the differences in bioavailability and toxicity possessed by different Se forms remains unknown and further studies are needed to better understand the critical metabolic processes that determine Se tolerance or toxicity. Based on the obtained results and considering that the inorganic forms of Se (selenite and selenate) have a higher solubility in water and are cheaper than organic forms (Se-Cys and Se-Met), we concluded that 20 mg kg^−1^ of Se from selenite can be suggested as the best combination of Se form and concentration suitable to in vitro inhibit the development of the considered *F. proliferatum* strain PG-CH1.

Even if the use of a single strain of *F. proliferatum* does not allow a general conclusion to be drawn at the species level, the results obtained may provide a basis for further investigations concerning Se effects against this polyphagous and global fungal pathogen, and for other *Fusarium* or fungal species. Further developments of this research can be undertaken by screening a higher number of *F. proliferatum* strains to assess the inhibitory activity of Se in a wider population context. This because not all isolates of a species may be sensitive to Se activity, and the phenomenon of Se tolerance by fungi has been previously described [61].

In addition, the activity of Se on the biosynthesis of fumonisins mycotoxins by *F. proliferatum* may also be interesting to explore, in addition to the impact of Se on the growth of other *Fusarium* species and their mycotoxins. Finally, these results may be also useful for evaluating the ability of Se to control *F. proliferatum* infections of rice in the field, in addition to this or other *Fusarium* species in different cereal hosts. Moreover, the addition of low concentrations of Se from selenite to conventional fungicides may be a promising alternative approach for the control of *Fusarium* species in different cereal hosts [62,63].

Many fungicides are categorized as polluting, and it can be reasonably assumed that the addition of Se may help in reducing the concentration of the chemical active ingredient by maintaining the same fungicidal efficacy and decreasing the potentially hazardous effect on the environment and human health [17,19] simultaneously realizing a Se-biofortification of cereal grains [64]. Within the U-shaped range [14], Se is an essential micronutrient with beneficial effects in animals and humans [65].

## 4. Materials and Methods

### 4.1. Chemicals and Reagents

Sodium selenite (Na_2_SeO_3_; selenite), sodium selenate (Na_2_SeO_4_; selenate), Se-L-Cystine (C_6_H_12_N_2_O_4_Se_2_; Se-Cys), Se-DL-methionine (C_5_H_11_NO_2_Se; Se-Met), Hoagland stock solution, NaCl, streptomycin sulfate, protease, ethanol 95%, agarose, trizma base-glacial acid acetic-ethylene diamine tetra acetic acid disodium salt dihydrate (TAE), EF1 and EF2 primers, glutaraldehyde, and phosphate buffer were all purchased from Sigma-Aldrich (Saint Louis, MO, USA). PDA was purchased from Biolife Italiana (Milan, Italy). Sodium hypochlorite (NaClO) 7% was purchased from Carlo Erba Reagents (Milan, Italy). RedSafe was purchased from iNtRON Biotechnology (Burlington, MA, USA). Gene Ruler1 kb was purchased from Thermo Fisher Scientific (Walthman, MA, USA). Dnase free sterile water was purchased from 5prime (Hilden, Germany). HyperLadder 100–1000 bp was purchased by Bioline (Cincinnati, OH, USA).

### 4.2. Obtainment of Fusarium proliferatum Strain PG-CH1 from Rice Seedlings

The *F. proliferatum* strain PG-CH1 used in this study was isolated from rice seedlings (variety Selenium). To assess the fungal microorganisms affecting rice seedlings, portions of symptomatic material were surface disinfected for 1 min with water-ethanol 95%-NaClO (7%, solution) (82:10:8% vol.) and rinsed with sterile water for 1 min. After the disinfection process, small pieces (0.5 cm) of seedling tissue were placed onto PDA supplemented with streptomycin sulfate (0.16 g L^−1^) into 5 Petri dishes (90 mm diameter) containing 7 pieces each (7 pieces from one single seedling, 1 seedling per plate, 5 seedlings in total). The dishes were incubated at 22 °C in the dark, and after 5 days a combination of visual and stereomicroscope (SZX9, Olympus, Tokyo, Japan) observations were carried out on each piece to assess fungal development. After visual observations, fungal colonies having similar morphology on PDA showed a higher (95%) isolation incidence.

According to colony color and shape on PDA from visual examination, in addition to the morphology of reproductive structures by microscope analysis (Axiophot, Zeiss, Oberkochen, Germany), they were considered to potentially belong to the genus *Fusarium* and a representative isolate (named PG-CH1) of all of those obtained was transferred onto new plates containing PDA and grown at 22 °C in the dark. After the obtainment of monosporic culture, the isolate was placed onto new PDA plates at 22 °C in the dark for two weeks. Successively, the mycelium was scraped from the PDA surface and placed into 2 mL sterile plastic tubes (Eppendorf, Hamburg, Germany) and stored at −80 °C. Following freeze drying with a lyophilizer (Heto Powder Dry LL3000; Thermo Fisher Scientific, Walthman, MA, USA), the mycelium was finely ground with a grinding machine (MM200, Retsch, Dusseldorf, Germany) for 6 min with a frequency of 25 Hz.

DNA extraction was carried out using the method previously described in [66]. Extracted genomic DNA was visualized on a 1% agarose, TAE gel in TAE buffer (1X) containing 500 μL L^−1^ of RedSafe. DNA fragments were separated in 10 cm long agarose gels, with an electrophoresis apparatus (Eppendorf) applying a tension of 110 V for ~30 min. Electrophoretic runs were visualized using an ultraviolet transilluminator (Euroclone, Milan, Italy). DNA concentration was estimated by comparison with Gene Ruler 1 kb included in each gel as a control. DNA was diluted with Dnase free sterile water for molecular biology use to obtain a concentration of ~30 ng μL^−1^ and stored at −20 °C until use. The DNA extracted from the *Fusarium* isolate was subjected to *tef1α* gene amplification, purification, and sequencing. PCR protocol is described in [67] adopting EF1 (ATGGGTAAGGA(A/G)GACAAGAC) and EF2 (GGA(G/A)GTACCAGT(G/C)ATCATGTT) primers [68]. PCR assays were performed on a T-100 thermal cycler (Bio-Rad, Hercules, CA, USA). The PCR fragment was visualized on TAE 1X agarose gel (2%) containing 500 μL L^−1^ of RedSafe. The DNA fragment was separated by an electrophoresis apparatus applying a tension of 110 V for ~40 min. Electrophoretic runs were observed with an ultraviolet transilluminator. The size of the amplified fragment was obtained by comparison with HyperLadder 100–1000 bp. The PCR fragment was purified and sequenced by an external sequencing service (Genewiz Genomics Europe, Takeley, UK). The sequence obtained was verified by Chromatogram Explorer Lite v 4.0.0 (HeracleBiosoft srl, Mioveni, Romania, 2011) and compared to those deposited on the BLAST database (National Center for Biotechnology Information (NCBI) Basic Local Search Tool (BLAST)), available online at http://blast.ncbi.nlm.nih.gov (accessed on 1 November 2018) [69], and the *Fusarium* MLST database (available online at http://fusarium.mycobank.org/, accessed on 1 November 2018).

To further confirm the species identity, the strain PG-CH1 was subjected to phylogenetic analysis based on *tef1α* sequences. The sequence of the PG-CH1 strain was aligned with 15 reference *tef1α* sequences of most common pathogens associated with seedling rice diseases such as *F. fujikuroi*, *F. verticillioides*, and *F. proliferatum* (Table 1) [36,70]. The sequences were aligned using the Muscle Algorithm implemented in the MEGA 7 software package [71]. Phylogenetic analysis was conducted in MEGA, via maximum likelihood following the best fit model of molecular evolution as determined by Bayesian information criterion (BIC) scores. A discrete Gamma distribution was used to model evolutionary rate differences among sites. All positions with less than 90% of coverage were eliminated. Statistical support of branches was evaluated using bootstrap analysis of 1000 replicates. The out-group isolate *F. oxysporum* MRC 1694 (accession number MH582350) was used for rooting the tree [72].

### 4.3. In Vitro Evaluation of the Inhibitory Activity of Se Forms on F. proliferatum Strain PG-CH1

PDA was supplemented with four different Se forms (selenite, selenate, Se-Cys, and Se-Met) at five different Se concentrations (5, 10, 15, 20, and 100 mg kg^−1^ of Se). A weighed amount of each Se form was dissolved in sterile distilled water to obtain a concentrated solution of 7 g L^−1^ of Se. A predetermined volume (64.3, 128.6, 192.8, 257.1, and 1280.0 μL) of the Se-concentrated obtained solutions was added to 20 g of PDA so that Se concentrations in the growth medium were 5, 10, 15, 20, and 100 mg kg^−1^. Successively, mycelium plugs (5 mm diam.) were taken from the edge of one week old *F. proliferatum* strain PG-CH1 colonies, developed on PDA at 22 °C, in the dark. The plugs were placed onto Se amended PDA, at the center of the plate, with the mycelium side facing upwards. Untreated controls (0 mg kg^−1^ of Se), and control treatments with NaCl (as counter ions of selenite and selenate), were also included in the experiment to ascribe the observed growth reduction of *F. proliferatum* strain PG-CH1 to Se with certainty. For this purpose, NaCl was added to the growth medium at the concentration of 254 mg kg^−1^ to fix the amount of Na^+^ at 100 mg kg^−1^, which is the same as the maximum Se concentration used in the experiment. Furthermore, the selected concentration of 254 mg kg^−1^ of NaCl (100 mg kg^−1^ of Na^+^) allowed the concentration of Na^+^ combined with inorganic Se (selenite and selenate) to be exceeded. After 10 days of incubation at 22 ± 2 °C in the dark, radial growth of fungal colonies was measured by ImageJ software (https://imagej.nih.gov/ij/, accessed on 20 January 2019), and the inhibitory activity was expressed as the percentage of radial growth reduction concerning the untreated control, according to Equation (1):(1)*Radial growth reduction of F. proliferatum strain PG−CH1colonies* (%) = (radial growth_control_ − radial growth_treatment_): radial growth_control_ × 100

Four replicates per treatment were performed. This experiment was repeated twice and the results of the second experiment, representative of the entire study, are shown. The preliminary first experiment showed a colony growth pattern, which was confirmed by the results obtained in the second experiment. The following analyses were realized exclusively using the materials of the second experiment.

### 4.4. Se Speciation Analysis

The Se speciation analysis was performed on 10 day old *F. proliferatum* colonies grown on amended PDA. About 16 g of sample (the sample comprises both mycelium and growth medium) was added with 10 mL of distilled water to 50 mL centrifuge tubes, accurately stirred to disperse the PDA gel, and sonicated for 2 min with an ultrasound probe. Then, protease was added up to 2.0 mg mL^−1^ and the obtained sample was stirred in a water bath at 37 °C for 15 h. Because PDA contained 2% by weight of agar, and considering that agar forms a gel in the concentration range of 0.5–2% by weight, the obtained samples were 10-fold diluted with distilled water, then cooled at room temperature and centrifuged at 5000 rpm for 10 min. Then, the supernatant was collected and filtered through 0.22 μm Millex GV filters (Millipore Corporation). The Se standards selenite, selenate, Se-Met, and Se-Cys were prepared in ultrapure (>18 MΩ) water (Appendix A). Speciation of Se was performed using a Liquid Chromatography-Inductively Coupled Plasma Mass Spectrometry (LC-ICP-MS/MS) system consisting of an Agilent 1260 Infinity II LC system and an Agilent 8900 ICP-tandem mass spectrometer (Agilent Technologies Japan, Ltd., Tokyo, Japan). Details of the mobile phases, column, and MS/MS conditions are available in Appendix A.

### 4.5. Observation by Scanning Electron Microscopy

For this analysis, based on the results obtained from in vitro evaluation of the inhibitory activity of different Se forms on *F. proliferatum* strain PG-CH1 growth, we focused our attention on Se activity from selenite. In detail, a mycelium plug (5 mm diameter) of *F. proliferatum* developed in the presence of 0, 5, 10, 15, 20, and 100 mg kg^−1^ of Se from selenite was sampled from 10 day old colonies to perform SEM analysis. A mycelium plug was collected from two replicates of the previously described experiment for a total of 12 samples (5 different Se concentrations from selenite plus the untreated control). Samples were prepared for SEM observations following the protocol described in [73] with slight modifications. Samples were fixed in 5% glutaraldehyde for 24 h; washed three times with 0.1 M phosphate buffer, pH 7.2; rinsed three times in distilled water; and dehydrated in ethanol series (25%, 50%, 75%, 90%, and 100%) for 7 min each. Samples were then transferred to a critical point dryer (Emitech K850, Quorum Technologies Ltd., Laughton, UK) to complete the drying process with carbon dioxide as a transition fluid. Specimens were then mounted on aluminum stubs with double-sided carbon tape, coated with gold in a sputter (Emitech K500, Quorum Technologies Ltd., Tokyo, Japan), and observed with a SEM 515 (Philips, Amsterdam, The Netherlands) at 12 kV. Observations were conducted for all 12 samples.

### 4.6. Statistical Analysis

Data of radial growth reduction of *F. proliferatum* strain PG-CH1 colonies after 10 days of incubation at 22 ± 2 °C in the dark in the presence of different Se concentrations from different Se forms were subject to one-way ANOVA. Data were analyzed considering the “radial growth reduction of *F. proliferatum* strain PG-CH1 colonies” (variable) for each “Se concentration” or “Se form” (factors). The results were expressed as the mean of four biological replicates (±standard error). To check for pairwise contrasts, Tukey Honestly Significant Difference multiple comparison tests were performed (*p* < 0.05) using the Microsoft Excel (Microsoft Corporation, Redmond, WA, USA) Macro “DSAASTAT ver. 1.0192” (macro developed by University of Perugia, Italy) [74].

## Figures and Tables

**Figure 1 plants-10-01725-f001:**
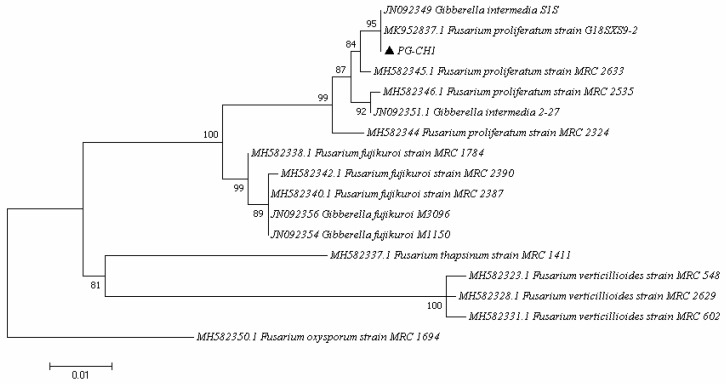
Phylogenetic relationship of PG-CH1 strain (triangle) and members of the *Fusarium fujikuroi* species complex shown in a maximum likelihood dendrogram based on the Kimura 2-parameter model. The tree with the highest log likelihood (–1567.09) is shown. Bootstrap values are indicated above the branch nodes. A discrete Gamma distribution was used to model evolutionary rate differences among sites.

**Figure 2 plants-10-01725-f002:**
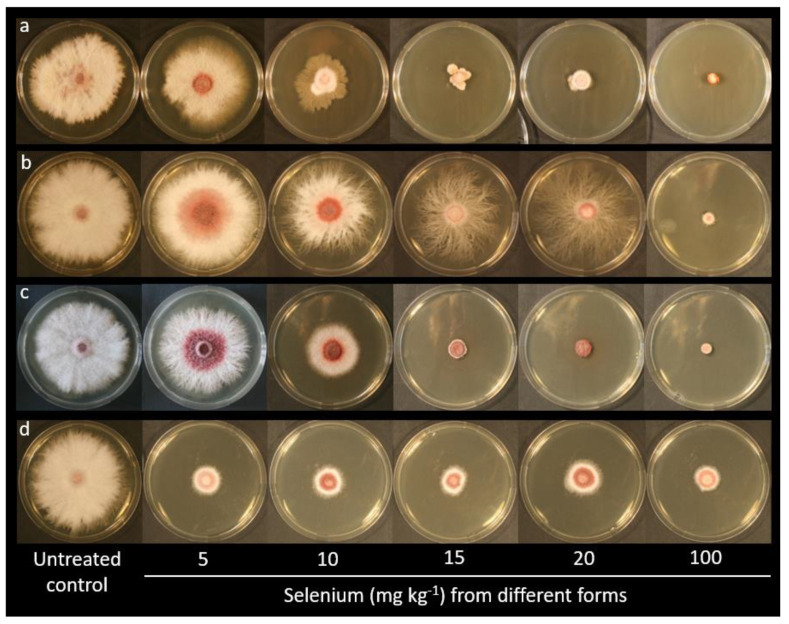
Effect of increasing selenium concentrations from different selenium forms on the colony development of *Fusarium proliferatum* strain PG–CH1 after 10 days of incubation at 22 ± 2 °C in the dark in comparison to the untreated control. Selenite (**a**), selenate (**b**), selenomethionine (**c**), selenocystine (**d**).

**Figure 3 plants-10-01725-f003:**
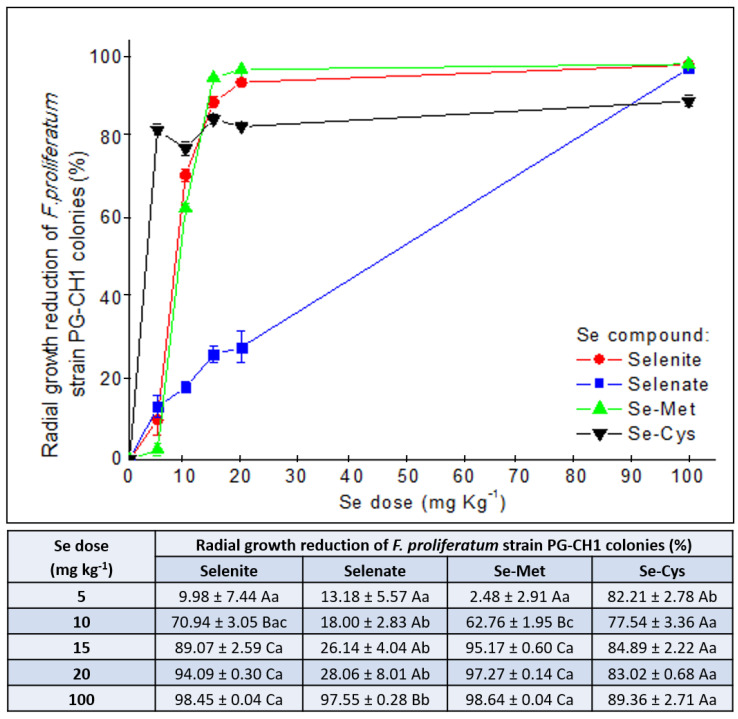
Effect of five selenium (Se) concentrations (5, 10, 15, 20, and 100 mg kg^−1^) from sodium selenite (selenite; red circles), sodium selenate (selenate; blue squares), selenomethionine (Se–Met; green up–triangles), and selenocystine (Se-Cys; black down-triangles) on *Fusarium proliferatum* strain PG-CH1 colony development. The inhibitory activity was measured after 10 days of incubation at 22 ± 2 °C in the dark and expressed as the mean (± standard error) of colony radial growth reduction (%) relative to the untreated control (0 mg kg^−1^ of Se), calculated according to the equation: (radial growth_control_ − radial growth_treatment_): radial growth_control_ × 100. One-way ANOVA was used to determine statistically significant differences in radial growth of the tested strain. In the table, within a Se form (A–C) or a Se concentration (a–b), means with the same letter are not significantly different at *p* < 0.05 based on the Tukey Honestly Significant Difference multiple comparison test.

**Figure 4 plants-10-01725-f004:**
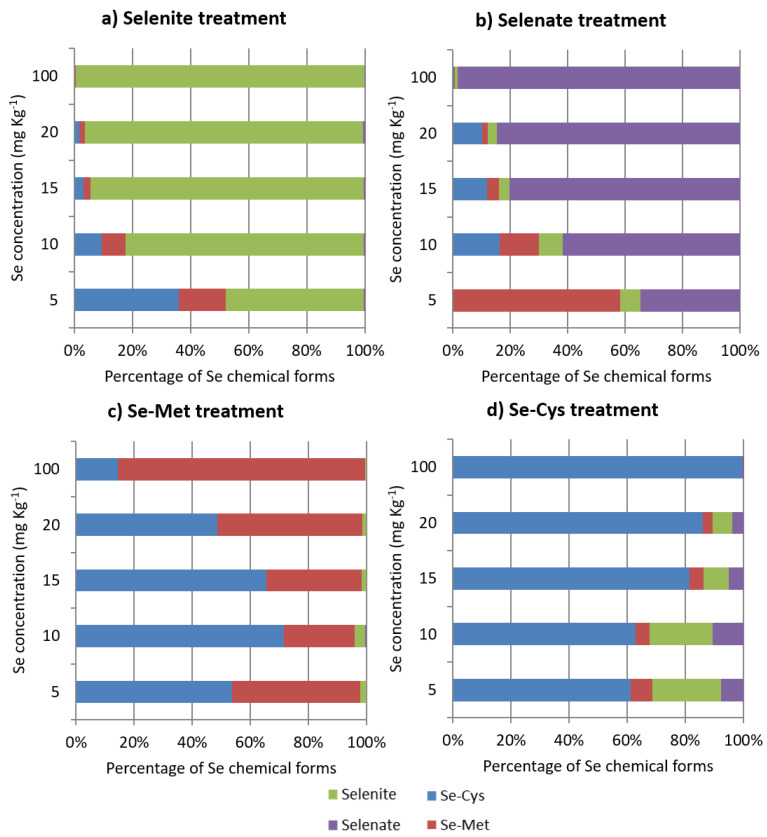
Selenium (Se) speciation data showing different Se chemical forms and their distribution after 10 days of incubation in the growth medium (potato dextrose agar) in the presence of *Fusarium proliferatum* strain PG-CH1 and following the application of increasing Se concentrations (5, 10, 15, 20, and 100 mg kg^−1^) as sodium selenite (selenite; **a**), sodium selenate (selenite; **b**), selenomethionine (Se-Met; **c**), and selenocystine (Se-Cys; **d**).

**Figure 5 plants-10-01725-f005:**
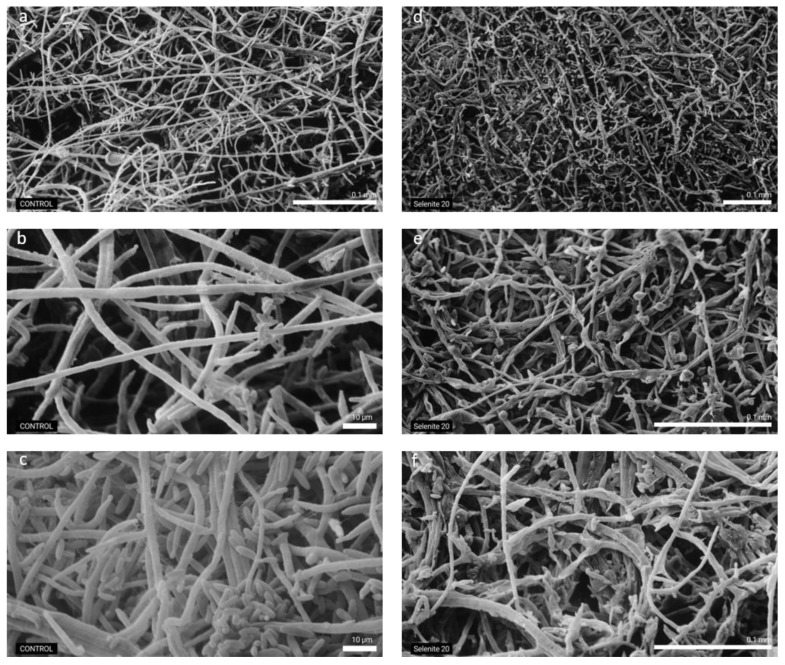
Scanning Electron Microscopy images showing *Fusarium proliferatum* strain PG-CH1 hyphae grown for 10 days on potato dextrose agar (PDA) not amended with selenium (Se) (labelled as control) (**a**–**c**) and on PDA amended with 20 mg kg^−1^ of Se as sodium selenite (selenite; labelled as selenite 20) (**d**–**f**). The effect of 20 mg kg^−1^ of Se as selenite caused appreciable modifications in the morphology and density of *F. proliferatum* strain PG-CH1 hyphae (**d**–**f**) in comparison to the untreated control (**a**–**c**).

**Table 1 plants-10-01725-t001:** *Fusarium* spp. strains used in the phylogenetic analysis and related information.

Strain	Species	Species Complex	Host	Geographic Origin/Substrate	GenBank Accession Number	References
MRC 548	*F. verticillioides*	*Fujikuroi*	Maize	South Africa	MH582323.1	[72]
MRC 602	*F. verticillioides*	*Fujikuroi*	Maize	South Africa	MH582331.1	[72]
MRC 1411	*F. tapsinum*	*Fujikuroi*	Maize	North Carolina-USA	MH582337.1	[72]
MRC 1784	*F. fujikuroi*	*Fujikuroi*	Rawcotton	Georgia, USA	MH582338.1	[72]
MRC 2324	*F. proliferaum*	*Fujikuroi*	Cotton boll	Alabama, USA	MH582344.1	[72]
MRC 2387	*F. fujikuroi*	*Fujikuroi*	Rice	Japan	MH582340.1	[70]
MRC 2390	*F. fujikuroi*	*Fujikuroi*	Unknown	Unknown	MH582342.1	[72]
MRC 2535	*F. proliferatum*	*Fujikuroi*	River sediment	Japan	MH582346.1	[72]
MRC 2629	*F. verticillioides*	*Fujikuroi*	Maize	Iowa, USA	MH582328.1	[72]
MRC 2633	*F. proliferatum*	*Fujikuroi*	Wheat	India	MH582345.1	[72]
S1S	*F. proliferatum*	*Fujikuroi*	Rice (seed)	Italy	JN092349	[70]
2–27	*F. proliferatum*	*Fujikuroi*	Rice (seed)	Italy	JN092351	[70]
M3096	*F. fujikuroi*	*Fujikuroi*	Rice	Georgia, USA	JN092356	[70]
M1150	*F. fujikuroi*	*Fujikuroi*	Rice	Taiwan	JN092354	[70]
G18SXS9-2	*F. proliferatum*	*Fujikuroi*	Unknown	Unknown	MK952837.1	Unknown
MRC 1694	*F. oxysporum*	*Oxysporum*	Unknown	Human	MH582350.1	[72]

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
