# Peer review of "In Vitro Evaluation of the Inhibitory Activity of Different Selenium Chemical Forms on the Growth of a Fusarium proliferatum Strain Isolated from Rice Seedlings"

_plants, 2021, doi:10.3390/plants10081725_

Round 1

Reviewer 1 Report

The study, entitled “In vitro Evaluation of the Inhibitory Activity of Different Sele-2 nium Chemical Forms on a Fusarium proliferatum Strain Iso-3 lated from Rice Seedlings ” tests different Se forms on the development of a Fusarium proliferatum strain, with the aim to select a least costly Se concentration as an alternative to toxic conventional fungicides used for the control of Fusarium species. In general, the study is well-designed, the experiments are documented with extensive controls, the manuscript is well written and the results are clearly presented and the drawn conclusions are valid.

Main comment:

The study mentions the relation of Se with oxidative stress (OS) in Introduction. However, the Discussion section lacks the presentation of a mechanism that would possibly explain in what developmental context Se-induced OS can act as an alternative fungicide. Such mechanism is already known and is related to metamorphosis (sclerotial) in filamentous fungi, which is induced by oxidative stress, while its inhibition by decreased OS causes inhibition of sclerotium formation and fungal propagation and death (DOI: https://doi.org/10.1093/icb/icj034). The extension of this theory predicts that any antioxidant (e.g., Se, acting indirectly as antioxidant by being cofactor of glutathione peroxidase) can stop fungal phytopathogen propagation by inhibiting its metamorphosis (sclerotial, conical), thus acting as natural fungicide.

Such sclerotial metamorphosis takes place in the fungi, mentioned in the study (in Introduction), where Se-related developmental effects have been studied. Examples for sclerotia formation are Aspergillus sp (DOI: https://doi.org/10.1016/j.funbio.2021.01.008), (DOI: https://doi.org/10.1371/journal.pone.0146169), Alternaria sp (https://agris.fao.org/agris-search/search.do?recordID=IT2001061757), Penicillium sp (https://doi.org/10.1016/j.jcis.2013.01.023), and Botrytis sp (DOI: https://doi.org/10.1016/S0007-1536(79)80089-3). In Aspergillus flavus, oxidative stress (glutathione peroxidase) is a prerequisite for aflatoxin production (DOI: 10.1016/s0891-5849(00)00398-1), and also in sclerotial differentiation (DOI: 10.1128/AEM.01282-14).

Discussing the results of the study in the context of such mechanism will strengthen its scientific validity, and provide thoughts for testing other antioxidants, besides Se, as alternatives for conventional phytopathogen fungicides.

Author Response

We are thankful to reviewer 1 to suggest an additional possible mechanism to explain the fungicide activity of Se. For this reason, we added the description of this mechanism in the discussion section as requested by reviewer 1. Please see the revised version of the manuscript (lines: 335-341). 

Reviewer 2 Report

The article: "In vitro Evaluation of the Inhibitory Activity of Different Selenium Chemical Forms on a Fusarium proliferatum Strain Isolated from Rice Seedlings" by E. Troni , G. Beccari, R. D’Amato, F. Tini, D. Baldo, M.T. Senatore, G.M. Beone, M.C. Fontanella, A. Prodi, D. Businelli and L. Covarelli, provides interesting data on the activity of various inorganic and organic forms of Selenium against a strain of Fusarium proliferatum isolated from rice and pathogen in various species of agricultural interest. The use of selenium salts (selenite) at low concentrations could show a good ability to limit this pathogen.
However, while the title mentions "inhibitory activity", the document only reports data on the in vitro growth of the pathogen colonies, but no data on the production and germination of the conidia of the pathogen are reported. It is therefore proposed to change the title as follows:
In vitro Evaluation of the Inhibitory Activity of Different Selenium Chemical Forms on the Growth of a Fusarium proliferatum Strain Isolated from Rice Seedlings. At the end of the Discussion, mention could also be made of the relationship between fungi and resistance to selenium.   In addition, there are some small typos:
in the Abstract, page 1, Fusarium proliferatum (lines 18, 29) and Fusarium (line 30) should be written in italics;
in the Introduction on page 2, the names of Fusarium species should all be abbreviated (F. graminearum, F. fujikuroi, etc.);
Results, p. 3, line 118 F. proliferatum (with a space);
Discussion, page 9, line 286, Fusarium spp (a p is in bold);
Materials and Methods, page 14, line 400, (variety Selenium) is a repetition;
row 407, (7 pieces per seedling, 5 seedlings) is one repeat;
page 12, line 451 package [70].

Author Response

REQUESTS OF REVIEWER 2 AND POINT BY POINT RESPONSE OF AUTHORS

  • However, while the title mentions "inhibitory activity", the document only reports data on the in vitro growth of the pathogen colonies, but no data on the production and germination of the conidia of the pathogen are reported. It is therefore proposed to change the title as follows: In vitro Evaluation of the Inhibitory Activity of Different Selenium Chemical Forms on the Growth of a Fusarium proliferatum Strain Isolated from Rice Seedlings.

RESPONSE: We agree with reviewer 2 and we changed the title of the manuscript. Please see in the revised version of the manuscript the new version of the title: “In vitro Evaluation of the Inhibitory Activity of Different Selenium Chemical Forms on the Growth of a Fusarium proliferatum Strain Isolated from Rice Seedlings” as suggested by reviewer 2 (lines 1-4).

  • At the end of the Discussion, mention could also be made of the relationship between fungi and resistance to selenium.

RESPONSE: We are thankful to reviewer 2 to bring to our attention this aspect. We mention at the end of the discussion the tolerance to Se by fungi that is certainly very important (lines 374-376).

  • in the Abstract, page 1, Fusarium proliferatum (lines 18, 29) and Fusarium (line 30) should be written in italics;

RESPONSE: Following the comments of reviewer 2 we checked the abstract and we realized that probably the PDF transformation process changed some words that were in italics. In addition, the abstract was deleted in the version of the manuscript that we downloaded for revision from the submission form. Please note that we added the abstract and in the word document of the revised manuscript scientific names are in italics (lines 16-30).

  • in the Introduction on page 2, the names of Fusarium species should all be abbreviated (F. graminearum, F. fujikuroi, etc.);

RESPONSE: We used the extension form of species name when we mention it for the first time even if the genus name has been already cited. This helps the reader to better understand the name of a species when it is mention for the first time. If reviewer 2 agree we leave the extended name of a species the first time that it is mention but if reviewer 2 prefers that they are abbreviated we can make it.

  • Results, p. 3, line 118 F. proliferatum (with a space);

RESPONSE: We added the space (line 116).

  • Discussion, page 9, line 286, Fusarium spp (a p is in bold);

RESPONSE: We corrected the “p” that was in bold (line 284).

  • Materials and Methods, page 14, line 400, (variety Selenium) is a repetition

RESPONSE: We removed the repetition (line 409).

  • row 407, (7 pieces per seedling, 5 seedlings) is one repeat

RESPONSE: We removed also this repetition (line 416).

  • page 12, line 451 package [70].

RESPONSE: We made the correction suggested by reviewer 2 (460).

We are very grateful to reviewer 2 for the valuable revision that certainly improves the quality of the paper.
